# Identification of Dermatophyte and Non-Dermatophyte Agents in Onychomycosis by PCR and DNA Sequencing—A Retrospective Comparison of Diagnostic Tools

**DOI:** 10.3390/jof8101019

**Published:** 2022-09-27

**Authors:** Isabella Pospischil, Charlotte Reinhardt, Olympia Bontems, Karine Salamin, Marina Fratti, Gabriela Blanchard, Yun-Tsan Chang, Helga Wagner, Philipp Hermann, Michel Monod, Wolfram Hoetzenecker, Emmanuella Guenova

**Affiliations:** 1Department of Dermatology, Kepler University Hospital, Johannes Kepler University Linz, 4020 Linz, Austria; 2Department of Dermatology, Lausanne University Hospital (CHUV) and the Faculty of Biology and Medicine, University of Lausanne, 1007 Lausanne, Switzerland; 3Center for Clinical Studies (CCS Linz), Johannes Kepler University Linz, 4020 Linz, Austria; 4Institute of Applied Statistics, Johannes Kepler University Linz, 4020 Linz, Austria; 5Department of Dermatology, University Hospital Zurich and the University of Zurich, 8091 Zurich, Switzerland; 6Department of Dermatology, Hospital 12 de Octubre, Medical School, University Complutense, 28040 Madrid, Spain

**Keywords:** dermatophyte, fungal culture, molecular mycology, NDM, onychomycosis, PCR, sequencing, tinea pedis, *Trichophyton*

## Abstract

Rapid and reliable fungal identification is crucial to delineate infectious diseases, and to establish appropriate treatment for onychomycosis. Compared to conventional diagnostic methods, molecular techniques are faster and feature higher accuracy in fungal identification. However, in current clinical practice, molecular mycology is not widely available, and its practical applicability is still under discussion. This study summarizes the results of 16,094 consecutive nail specimens with clinical suspicion of onychomycosis. We performed PCR/sequencing on all primary nail specimens for which conventional mycological diagnostics remained inconclusive. In specimens with a positive direct microscopy but negative or contaminated culture, molecular mycology proved superior and specified a fungal agent in 65% (587/898). In 75% (443/587), the identified pathogen was a dermatophyte. Positive cultures for dermatophytes, yeasts and non-dermatophyte molds (NDMs) were concordant with primary-specimen-DNA PCR/sequencing in 83% (10/12), 34% (22/65) and 45% (76/169), respectively. Among NDMs, agreement was high for *Fusarium* spp. (32/40; 80%), but low for *Penicillium* spp. (5/25; 20%) and *Alternaria* spp. (1/20; 5%). This study underlines the improvement in diagnostic yield by fungal primary-specimen-DNA PCR/sequencing in the event of a negative or contaminated culture, as well as its significance for the diagnosis of dermatophyte and non-dermatophyte onychomycosis. Molecular mycology methods like PCR and DNA sequencing should complement conventional diagnostics in cases of equivocal findings, suspected NDM onychomycosis or treatment-resistant nail pathologies.

## 1. Introduction

Onychomycosis is the most common cause of abnormal-appearing nails with an estimated mean prevalence of 4–9% in the general population [1,2,3]. The causative pathogens include dermatophytes, yeasts, and non-dermatophyte molds (NDM) with dermatophytes accounting for the majority of cases [2,4]. Involvement of NDMs in nail infection can be concluded with some confidence in the event of their repeated isolation from the affected nails [5,6,7].

Systemic antifungal drugs, such as terbinafine and azoles, are frequently necessary for the successful treatment of dermatophyte onychomycosis [8]. As these agents feature a risk of drug interactions and potential adverse effects, the indication for their administration needs to be carefully evaluated [9]. Furthermore, although typical clinical features can be highly suggestive of a fungal infection, several other diseases can closely resemble onychomycosis such as psoriasis vulgaris, lichen planus, trauma, and malignancies. Overall, these mostly noninfectious disorders account for 40–50% of all nail abnormalities and must be differentiated from onychomycosis, as they require fundamentally different treatment [2,10]. In the case of onychomycosis, detection of the causative fungus facilitates identification of the infection source and increases patient compliance with long-term therapy with potential side effects. Most importantly, the selection of the appropriate antimycotic drug depends on the causative fungal pathogen [8,11,12]. In particular, NDM onychomycoses are recalcitrant to terbinafine and azoles, and treatment by topical amphotericin B should be considered as first-line therapy [13,14,15]. Drug resistance is also an increasingly recognized concern with some dermatophytes, especially *Trichophyton rubrum* and *Trichophyton mentagrophytes* [16,17,18,19,20].

For the above-mentioned reasons, accurate and reliable diagnostic tools for the identification of the pathogenic fungi in nail infections are highly needed. Conventional methods include potassium hydroxide (KOH) preparation, histopathology and fungal culture, but these approaches have limitations. Direct microscopy of KOH preparations deliver immediate results; however, drawing conclusions regarding viability or fungal identification at the species level is not possible. Additionally, its diagnostic precision largely depends on the experience of the performing expert, resulting in a wide-ranging sensitivity of 67–93% [4,21]. The sensitivity of direct microscopy can be increased by using fluorescent staining such as Calcofluor white or Blankophor [21]. Fungal culture allows the speciation of the causative fungal agent and confirmation of its viability; however, results are rarely available before 2–3 weeks of culture time. Moreover, with an estimated sensitivity of 60%, the fungal culture technique bears a high risk of false-negative results [4,21]. Histopathological examination with periodic acid–Schiff (PAS) staining yields a comparatively high sensitivity, but, like KOH preparation, assessment of viability or species determination of the fungi is not possible [4,21].

To overcome these limitations, molecular methods are increasingly used for routine fungal diagnostics given their high sensitivity and rapid identification of the fungi within a day [22]. Different molecular mycological techniques are available and they mainly differ by the DNA extraction method and PCR primers used, as well as in the method of analyzing the PCR products [23]. Primers used were either specific for the detection of only one dermatophyte species [24,25,26,27], pan-dermatophyte primers (dermatophyte-specific primers to detect any but only dermatophyte species) [25] and/or pan-fungal primers (primers to detect any fungal species) [26]. Possible primer targets include ITS-5.8S and 28S rDNA [25,26], as well as genes encoding topoisomerase II [27] and chitin synthase [24,25]. Most described methods focused on dermatophytes and only a few paid attention to yeasts and NDMs as possible infectious agents in onychomycosis [23].

However, these modern methods also have limitations, for example, they cannot assess fungal viability and are comparatively cost-intensive. Therefore, they are not widely available for all practitioners confronted with suspected onychomycosis and their significance in clinical practice is still controversial. The main objective of this retrospective evaluation was to compare the results of conventional methods (direct fluorescence microscopy, fungal culture) with the results from PCR/sequencing of fungal DNA extracted directly from the primary nail specimen in a large cohort of suspected onychomycosis in a specialized mycology laboratory. The aim of the analysis was to assess the significance of the obtained results in clinical practice.

## 2. Materials and Methods

### 2.1. Study Design and Data Acquisition

All consecutive nail samples from patients with clinically suspected onychomycosis, referred for diagnostic analysis to the specialized dermatomycology laboratory of the Department of Dermatology at the University Hospital in Lausanne from 2013 to 2018, were included in this retrospective cohort study. Patient data were collected in agreement with the VITA Certified Dermatology Biobank (CHUV-2103-12). The analysis of the data set was performed at the Department of Dermatology at the Kepler University Hospital Linz. The study protocol was approved by the Johannes Kepler University Ethics Committee (ethics committee number 1298/2020).

### 2.2. Diagnostic Approach

Firstly, all nail samples were examined by direct fluorescence microscopy. For this purpose, a portion of each nail sample was examined in a dissolving solution containing a fluorochrome [28,29]. The solution was prepared by dissolving 1 g of sodium sulfide (Na2S) (Sigma, St. Louis, MO, USA) in 7.5 mL of distilled water and then adding 2.5 mL of ethanol [29]. Then 20 μL of a 1% aqueous solution of Tinopal UNPA-GX (fluorescent bleach 28; Sigma) was added to this mixture. The sample preparations were examined using a Zeiss Axioskop fluorescence microscope with excitation between 400 and 440 nm (Zeiss, Thornwood, NY, USA).

Additionally, two fungal cultures were obtained from each sample using a Sabouraud dextrose agar plate with chloramphenicol (50 μg/mL) and chloramphenicol plus cycloheximide (400 μg/mL). After incubation at 30 °C for 14–21 days, fungi were identified following macroscopic and microscopic examination [4,30,31].

If prior diagnostics with direct fluorescence microscopy and fungal culture were inconclusive or inconsistent with clinical findings, the infectious fungus species was identified from the primary nail specimens by a PCR/sequencing assay, as previously described [26], as long as some nail material remained for further analysis. During the culture time of 2–3 weeks, the remaining nail was stored unprocessed in a separate box at room temperature.

#### 2.2.1. Fungal DNA Extraction

Nail fragments (20 to 100 mg) were incubated overnight in 500 μL of sodium sulfide dissolving solution (10% [wt/vol] Na2S (Sigma), 25% [vol/vol] ethanol) without fluorochrome. After centrifugation at 8000× *g* for 2 min, the sample precipitate containing fungal elements was washed twice with distilled water [26,28]. Fungal DNA was extracted from primary nail samples using the DNeasy Plant Minikit (Cat. No 69104, Qiagen AG, Hombrechtikon, Switzerland) according to the manufacturer’s protocol.

#### 2.2.2. 28S rDNA Amplification

28S ribosomal DNA (rDNA) was amplified by PCR using ReadyMix Taq PCR Mix with MgCl_2_ (cat. n° P4600, Sigma) coupled with large-subunit of forward fluorescently labelled primer LSU1 (5′-GATAGCGMACAAGTAGAGTG-3′) and reverse primer LSU2 (5′-GTCCGTGTTTCAAGACGGG-3′) (Microsynth AG, Balgach, Switzerland) [32]. Extracted fungal DNA (5 μL), 1 μM (each) forward and reverse primers, and 25 μL of DNA polymerase reaction mixture were mixed with nuclease-free water to give a total reaction volume of 50 μL. The reaction mixture was incubated for 1 min at 94 °C; subjected to 30 cycles of 0.5 min at 94 °C, 0.5 min at 55 °C, and 0.5 min at 72 °C; and finally incubated for 10 min at 72 °C in an ABI 2720 thermocycler (Applied Biosystems, Inc., Carlsbad, CA, USA). Concentrations of the PCR products from the nail samples were estimated in 0.8% (wt/vol) agarose gels and the amount determined using DNA Molecular Weight Marker XIV (Roche) and ranged from no detection to 150 ng/μL.

#### 2.2.3. Species Identification by DNA Sequencing

DNA sequence analysis of the amplified 28S rDNA was performed for species identification (Microsynth AG (Balgach, Switzerland) on an FLX Genome Sequencer (454 Sequencing; Roche, Basel, Switzerland). The fungal species was determined by comparing the sequence obtained with those in the NCBI database. The data were collected and stored in a MOLIS laboratory information system (CompuGroup Medical [CGM] Lab Belgium NV, Barchon, Belgium).

### 2.3. Statistics

Absolute and relative frequencies are provided for nominal variables. Median and interquartile range (IQR) are computed for metric variables. To investigate the relation between culture and PCR/sequencing results, the absolute frequencies and conditional frequencies of the culture results conditioned on the PCR/sequencing results are reported. These distributions are visualized with bar charts.

## 3. Results

Statistical analyses were obtained for two main groups: For the whole study population (*n* = 16,094) and subset of specimens, which in addition to the standard conventional methods (direct fluorescence microscopy and fungal culture) also underwent primary-specimen-DNA PCR/sequencing in parallel. The latter will be referred to as the “PCR/sequencing group” (*n* = 1148). Baseline characteristics of the two groups are presented in Table 1.

### 3.1. Whole Study Population

In a total of 16,094 nail specimens with suspected onychomycosis, fungal elements were detected in 68.1% by fluorescence microscopy and 28.7% by fungal culture (Table 1). 

Both methods were positive in 4,441 (27.6%) and negative in 4,951 (30.8%) cases, resulting in an overall concordance of 58.4%. The majority of discordant results resulted from samples with positive direct microscopy, but negative fungal culture: overall, 59.5% of the specimens with detection of fungal elements by fluorescence microscopy showed a negative fungal culture.

Among the patients with a positive culture, dermatophytes were the most commonly identified fungi (47.9%) with *Trichophyton rubrum* being the most prevalent, followed by NDMs (29.1%), yeasts (21.8%) and mixed infections (1.2%).

### 3.2. PCR/Sequencing Group

In 1148 cases, the fungal standard diagnostic with fluorescence microscopy and fungal culture was insufficient or inconclusive and, therefore, was complemented by sequencing of the amplified fungal DNA extracted from the primary nail specimen. Given the high specialization and expertise of the mycology laboratory in Lausanne, fungal elements were detected in 99.7% of these samples by fluorescence microscopy. While fungal culture remained of low sensitivity and only detected a fungal pathogen in 21.8%, molecular mycology clearly outperformed it as PCR/sequencing yielded positive results in 63.5% of the cases (Table 1).

In the PCR/sequencing group, the positive conventional culture results were distributed as follows: NDMs 67.6%, yeasts 26%, dermatophytes 4.8% and mixed infections 1.6%. Molecular mycology results in the same group differed as dermatophytes were the most prevalent (63.5%), followed by NDMs (31.1%) and yeasts (5.3%). *Fusarium* spp. were the most prevalent NDM detected by PCR/sequencing and accounted for 43.2% (98/227) of NDM cases. The distribution and comparison of culture and PCR/sequencing results are shown in Table 2.

PCR/sequencing detected a total of 463 dermatophyte infections. Of these, only 2.2% were also identified as dermatophyte infections by fungal culture, whereas without molecular mycology, the majority of clinical specimens were considered negative/contaminated (95.7%; Table 2, Figure 1). By contrast, among yeast and NDM infections diagnosed by PCR/sequencing, the culture obtained concordant results with molecular mycology in 56.4% and 40.1%, respectively (Table 2, Figure 1).

When conventional fungal culture and molecular mycology (primary-specimens-DNA PCR/sequencing) were compared based only on positive or negative/contaminated results, there was a match in 48.9% of the cases. Concordance decreased to 26% when agreement to the species level was requested (Table 3).

Of interest, only 40.5% (190/469) of all negative cultures also showed a negative PCR/sequencing result (Table 3, Figure 2). In 59.5%, PCR/sequencing could identify a fungal agent and a significant proportion of those were dermatophyte infections: dermatophytes 44% (*Trichophyton rubrum* 37%, *Trichophyton interdigitale* 7%), NDMs 14% and yeasts 1%. Additionally, in cultures contaminated by fungi or bacteria, molecular mycology still identified a causative fungal agent in 72% (dermatophytes 55%, NDMs 15%, yeasts 2%). Overall, PCR/sequencing could disclose a fungal agent in 65% (587/898) of specimens with negative or contaminated culture and 75% (443/587) of these were dermatophytes.

Depending on the fungus identified in the culture, concordance with the corresponding PCR/sequencing result differed. While agreement was high (≥80%) when the culture was positive for *Trichophyton rubrum* (83%) or *Fusarium* spp. (80%), it was low (<50%) when *Aspergillus* spp., *Acremonium* spp., *Candida parapsilosis*, *Penicillium* spp. or *Alternaria* spp. was detected (Table 3, Figure 2). Overall, the cultural result was confirmed by subsequent PCR/sequencing at the species level in 83% (10/12), 45% (76/169) and 34% (22/65) of cultures positive for dermatophytes, NDMs and yeasts, respectively.

## 4. Discussion

This large-scale study demonstrated the improvement in diagnostic yield in onychomycosis by molecular diagnostics like primary-specimen-DNA PCR/sequencing. PCR/sequencing could confirm the suspected onychomycosis and identify the causative fungi in a high number of cases despite inconclusive standard diagnostics, especially in terms of a negative or contaminated fungal culture. The concordance rate at the species level between the cultural and corresponding PCR/sequencing results differed among the culturally detected fungi.

Regarding epidemiological findings, *Trichophyton rubrum* and dermatophytes in general were the most prevalent fungal agents in our whole study population, which is well in line with the currently available study data [1,33,34,35]. As fungal DNA PCR/sequencing was confined to cases with inconclusive standard diagnostics, mostly in terms of a positive direct microscopic but negative or equivocal cultural result, NDMs were the most prevalent fungi identified by culture in the PCR/sequencing group. However, after the diagnostics were complemented by PCR/sequencing, the majority of cases in this group were ascribed to dermatophytes, followed by NDMs and yeasts. Similar to recent studies [28,36], NDMs accounted for approximately 30% of the causative fungal agents in our data, supporting the reported increase in NDM onychomycoses compared with that in older studies [2,15]. *Fusarium* spp. was the most prevalent NDM identified by PCR/sequencing and its total number reached 21% of that of dermatophytes. Compared to previous data, in which *Fusarium* spp. accounted for 15% of the number of dermatophytes, this also indicates a continued increase in *Fusarium* spp. onychomycoses, which can be associated with poor treatment response [12,13,28,37].

If fungal culture remained sterile despite positive direct microscopy and/or highly suggestive clinical findings, PCR/sequencing could eventually identify a fungal agent in 59.5% of the cases. Upon fungal culture contamination with positive direct identification and/or evocative clinical manifestations, PCR/sequencing detected a causative fungus in 72%. Similarly, other authors reported detection of fungi by molecular methods in up to 91.3% of samples with negative or contaminated culture [25,34,36,38]. This is in line with the generally reported higher sensitivity of molecular methods (80–95% [21,39]) compared with those of fungal culture (31–59%) [4,21]) [40,41,42,43].

However, differences between diverse molecular techniques in sensitivity and specificity as well as in their ability to detect NDMs, yeasts and mixed infections need to be considered in clinical practice [19,34]. Accordingly, besides inadequate sample collection, technical issues also might have contributed to the lower frequency of positive PCR/sequencing results (63.5%) in specimens with positive direct microscopic examination in our study compared with that of previous studies using other molecular methods [28,36,38,43]. For example, sequencing cannot adequately detect mixed infections, which are estimated to account for approximately 10% of onychomycoses [28,36]. In this study, the PCR primer target was the 28S rDNA, which is less efficient compared with ITS-5.8S amplification in the discrimination of fungi to the species level. However, we found previously that 28S rDNA amplification is more sensitive and reliably identifies species at the genus level [26,31,32,36]. Therefore, it sufficiently discriminates tinea unguium and mold onychomycosis, which in onychomycosis is the crucial information needed to choose the proper treatment. Furthermore, fungal identification in nails by 28S rDNA amplification has already been tested on a large number of samples and therefore it can be considered as a proper primer target in onychomycosis [36]. Additionally, it needs to be considered that molecular methods were primarily studied for dermatophyte infections [34,38,42,43,44,45] and to a significantly lesser extent for NDM and yeast infections [28,36,41,46]. As in our data, molecular techniques only missed a minority of culture-positive dermatophyte infections, whereas they detected a large number of additional, culture-negative ones in previous studies [28,34]. However, study data for molecular methods detecting NDM and yeast nail infections is comparatively scarce, and sensitivity might be lower [28,41].

The current study elucidates the significance of PCR and fungal DNA sequencing in the diagnosis of NDM nail infections, which is known as challenging regarding the discrimination of actual NDM nail infection and frequently occurring contamination of fungal cultures by NDMs. A positive cultural result for an NDM was confirmed by subsequent PCR/sequencing in merely 45% of cases, whereas 41% showed a negative result. In contrast, among cultures diagnosed as contaminated by NDMs, fungal sequencing detected NDMs in 16% and, therefore, might indicate actual NDM nail infection rather than contamination. Additionally, PCR/sequencing detected NDMs in 14% of negative cultures. Similarly, one study reported poor agreement for culture and PCR for the detection of NDMs with single samples in 167 patients with suspected onychomycosis [46].

In our study, the concordance rate of culture and corresponding PCR/sequencing results differed depending on the culturally identified fungus. Positive cultures for *Trichophyton* spp., *Fusarium* spp. and *Scopulariopsis* spp. were frequently (83, 80 and 62%, respectively) confirmed by PCR/sequencing, whereas this was rarely the case for *Alternaria* spp. (5%) or *Pencillium* spp. (20%). Similar agreement rates were reported in previous studies [28,36]. For example, a study [36] showed agreement of PCR-restriction fragment length polymorphism (PCR-RFLP) with prior cultural results as follows: *Trichophyton* spp. 93.8%, *Fusarium* spp. 84%, *Scopulariopsis* spp. 50%, *Aspergillus* spp. 46.4%, *Acremonium* spp. 80%, *Penicillium* spp. 18%, *Candida* spp. 59% and *Alternaria* spp. 5%. It is reasonable to assume that concordant NDM detection by culture and PCR/sequencing indicates actual NDM nail infection; nevertheless, repeated detection remains the gold standard in the diagnosis of NDM onychomycosis [46]. However, according to our data and previous studies, a positive culture for *Trichophyton rubrum* or *Fusarium* spp. increases the likelihood of a clinically significant result.

As a limitation of the recent study, the missing information on antifungal pretreatment must be considered, as recent application of antifungals significantly decreases the sensitivity of fungal culture but to a lesser extent with molecular methods [42]. In addition, this comparison of diagnostic methods is limited by the fact that, given the retrospective nature of the study, all three methods were not simultaneously applied at the beginning of the diagnostic process, but PCR/sequencing was confined to cases with inconclusive prior standard diagnostics.

In conclusion, the presented data shows that especially in patients with clinically suspected onychomycosis but equivocal standard diagnostics, molecular methods can significantly contribute to the correct diagnosis. This finding is important as confirming the diagnosis of onychomycosis is essential to establish correct treatment and increases patient compliance for a commonly months-lasting therapy. Given the high prevalence of onychomycosis in the general population, molecular methods might not be currently available in the first place for all those confronted with onychomycosis. Furthermore, the advantages obtained with PCR methods for routine analysis must be weighed against the price of molecular biology reagents and equipment (costs for PCR/sequencing are approximately tenfold higher compared with those of conventional culture), and the expenditure of time required from trained personnel to perform the laboratory procedures. However, clinicians should be aware of the high risk of a false negative and contaminated fungal culture; and molecular methods should be added when conventional diagnostics are inconclusive, NDM onychomycosis is suspected or in the case of a treatment-resistant course of disease.

## Figures and Tables

**Figure 1 jof-08-01019-f001:**
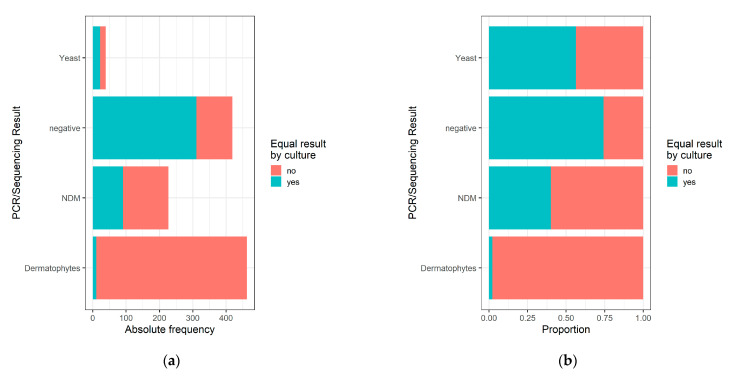
Comparison of PCR/sequencing with the respective culture results in absolute (**a**) and relative (**b**) numbers. Only a small number of dermatophyte infections identified by PCR/sequencing was also detected by culture, whereas the proportion was higher for yeast and NDM infections.

**Figure 2 jof-08-01019-f002:**
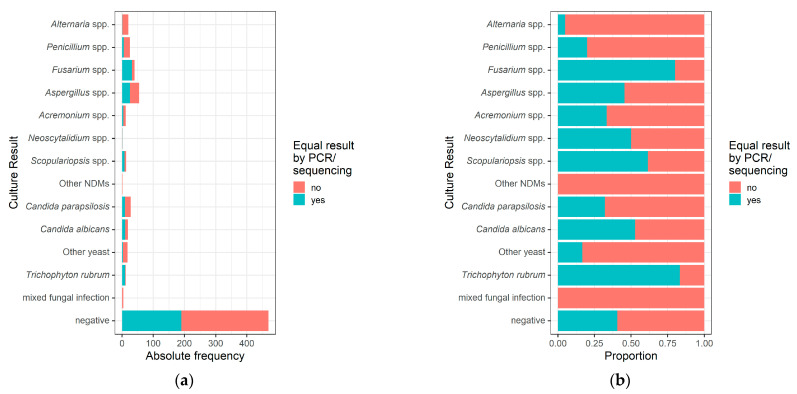
Comparison of the culture with the corresponding PCR/sequencing results in absolute (**a**) and relative (**b**) numbers. Agreement of culture with the subsequent PCR/sequencing results differed among the culturally identified fungi.

**Table 1 jof-08-01019-t001:** Characteristics of the whole study population and PCR/sequencing group.

		Whole Study Population *n* = 16,094	PCR/Sequencing Group *n* = 1148
Age Median years (IQR ^1^)		53 (23)	54 (21)
Sex n (%)	Female	10,066 (62.5%)	668 (58.2%)
	Male	6028 (37.5%)	480 (41.8%)
Sampling site N ^3^ (%)	Toenails	12,170 (75.6%)	925 (80.6%)
	Fingernails	1118 (7%)	74 (6.4%)
	Nail NOS ^2^	2745 (17.1%)	147 (12.8%)
	Subungual mass	57 (0.4%)	1 (0.1%)
Detection of fungal elements by diagnostic methods n (%)	Positive Microscopy	10,967 (68.1%)	1145 (99.7%)
	Positive Culture	4617 (28.7%)	250 (21.8%)
	Positive Sequencing	-	729 (63.5%)

^1^ Interquartile range. ^2^ Not otherwise specified. ^3^ 4 missing sampling sites in the whole study population and 1 missing in the PCR/Sequencing group.

**Table 2 jof-08-01019-t002:** Comparison of culture and PCR/sequencing results from the same primary nail specimen in 1148 patients.

		PCR/Sequencing Results			
		Dermatophyte	Yeast	NDM	Negative/Contamination
**Culture** **Result**	Dermatophyte *n* = 12	10 (2.2%)	0 (0%)	1 (0.4%)	1 (0.2%)
	Yeast *n* = 65	4 (0.9%)	22 (56.4%)	5 (2.2%)	34 (8.1%)
	NDM *n* = 169	5 (1.1%)	3 (7.7%)	91 (40.1%)	70 (16.7%)
	Mixed infection *n* = 4	1 (0.2%)	0 (0%)	0 (0%)	3 (0.7%)
	Negative/Contamination *n* = 898	443 (95.7%)	14 (35.9%)	130 (57.3%)	311 (74.2%)
	Total *n* = 1148	463 (100%)	39 (100%)	227 (100%)	419 (100%)

**Table 3 jof-08-01019-t003:** Comparison of culture results with their corresponding PCR/sequencing result.

Culture Result		Comparison With PCR/Sequencing				
		Equal Result	Discordant Result			
			NDM	Dermatophyte	Yeast	Negative
NDM *n* = 169	*Aspergillus* spp. *n* = 55	25 (45%)	2 (4%)	1 (2%)	2 (4%)	25 (45%)
	*Fusarium* spp. *n* = 40	32 (80%)	0 (0%)	3 (8%)	0 (0%)	5 (12%)
	*Penicillium* spp. *n* = 25	5 (20%)	6 (24%)	0 (0%)	1 (4%)	13 (52%)
	*Alternaria* spp. *n* = 20	1 (5%)	5 (25%)	0 (0%)	0 (0%)	14 (70%)
	*Scopulariopsis* spp. *n* = 13	8 (62%)	0 (0%)	0 (0%)	0 (0%)	5 (38%)
	*Acremonium* spp. *n* = 12	4 (33%)	3 (25%)	0 (0%)	0 (0%)	5 (42%)
	*Neoscytalidium* spp. *n* = 2	1 (50%)	0 (0%)	0 (0%)	0 (0%)	1 (50%)
	Other NDMs *n* = 2	0 (0%)	0 (0%)	0 (0%)	0 (0%)	2 (100%)
Yeast *n* = 65	*Candida parapsilosis* *n* = 28	9 (32%)	3 (11%)	1 (4%)	0 (0%)	15 (54%)
	*Candida albicans* *n* = 19	10 (53%)	0 (0%)	1 (5%)	0 (0%)	8 (42%)
	Other yeast *n* = 18	3 (17%)	2 (11%)	2 (11%)	0 (0%)	11 (61%)
Dermatophytes *n* = 12	*Trichophyton rubrum* *n* = 12	10 (83%)	1 (8%)	0 (0%)	0 (0%)	1 (8%)
Mixed fungal infection (NDM + yeast) *n* = 4		0 (0%)	0 (0%)	1 (25%)	0 (0%)	3 (75%)
Contamination (NDM) *n* = 149		0 (0%)	24 (16%)	101 (68%)	6 (4%)	18 (12%)
Contamination (yeast) *n* = 15		0 (0%)	0 (0%)	15 (100%)	0 (0%)	0 (0%)
Contamination (bacteria) *n* = 265		0 (0%)	39 (15%)	120 (45%)	3 (1%)	103 (39%)
Negative *n* = 469		190 (41%)	67 (14%)	207 (44%)	5 (1%)	0 (0%)
Total *n* = 1148		298 (26%)	152 (13%)	452 (39%)	17 (2%)	229 (20%)

## Data Availability

Authors confirm that all relevant data are included within this article.

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
