# Peer review of "Identification of Dermatophyte and Non-Dermatophyte Agents in Onychomycosis by PCR and DNA Sequencing—A Retrospective Comparison of Diagnostic Tools"

_jof, 2022, doi:10.3390/jof8101019_

Round 1

Reviewer 1 Report

The MS by Pospischil and co-workers is of most importance to the correct identification of dermatophytes and other fungi involved in nail infections, clearly indicating a strategy to improve the diagnosis of dermatophytosis but also to gather better epidemiological data relevant to health in general and specifically to public health. The manuscript is well written and the results are clearly transmitted. 

However there are some issues that can be improved most of which are related to the fact that the authors used as strategy to identify the agents the PCR of the 28S rDNA using the LSU1 and LSU2 primers, when in most of the studies in this area of medical mycology authors use the ITS-5.8S rDNA. This needs to be introduced and discussed because it will impact in the overall quality of the manuscript. 

Introduction:

Briefly describe the major molecular methods that have been used to identify dermatophytes and introduce some references in complement and more recent besides ref. 19. 

Methods (Line 103):

Please indicate the fluorescence microscopy method used. Even though the authors introduce a reference (20), it is important to introduce brief details about the method as done with the description of the culture methodology.

Results:

Table 2 is difficult to understand. Please describe the meaning of these results. These are not the same as in Figure 1?

Discussion:

Clearly, the discussion needs to be improved by comparing the results with similar studies, either using 28S or using ITSs, in dermatophytes and in other fungal groups. Since the strategy used by the authors was to amplify and by sequencing 28S directly from the nail sample it should be indicated if using either 28S or ITS the results are different.

 Moreover, lately there were studies using molecular biology approaches showing that sometimes the study of the rDNA is not sufficient to accurately identify dermatophytes (e.g T. rubrum complex identification). This should be included in the discussion. 

Minor comments:

Line 164 – 166: Italics?

Line 411: Reference 40?     

Author Response

Response to Reviewer 1 Comments

The MS by Pospischil and co-workers is of most importance to the correct identification of dermatophytes and other fungi involved in nail infections, clearly indicating a strategy to improve the diagnosis of dermatophytosis but also to gather better epidemiological data relevant to health in general and specifically to public health. The manuscript is well written and the results are clearly transmitted. 

However there are some issues that can be improved most of which are related to the fact that the authors used as strategy to identify the agents the PCR of the 28S rDNA using the LSU1 and LSU2 primers, when in most of the studies in this area of medical mycology authors use the ITS-5.8S rDNA. This needs to be introduced and discussed because it will impact in the overall quality of the manuscript. 

Response: We would like to thank the reviewer for his positive feedback. We have carefully revised the manuscript according to his insightful remarks and provided point-by-point responses to his comments.

Comment 1.1: Introduction:

Briefly describe the major molecular methods that have been used to identify dermatophytes and introduce some references in complement and more recent besides ref. 19

Response 1.1: The methods of molecular mycology have been nicely and comprehensively reviewed by Verrier and Monod, 2016 (doi: 10.1007/s11046-016-0038-z). A brief overview is now given in the revised manuscript (Line 83 – 94)

Comment 1.2: Methods (Line 103):

Please indicate the fluorescence microscopy method used. Even though the authors introduce a reference (20), it is important to introduce brief details about the method as done with the description of the culture methodology.

Response 1.2.: Thank you for this remark. We agree and a brief description of the method was included in the revised manuscript (Line 115 – 122)

Comment 1.3: Results:

Table 2 is difficult to understand. Please describe the meaning of these results. These are not the same as in Figure 1?

Response 1.3:  Table 2 represents a cross table comparing the results from PCR/sequencing and fungal culture from the same nail specimen in the 1148 patients from the “PCR/sequencing group”. It aims to give the reader a quick overview of the data and especially of the concordance/discordance of the results with these two methods. For example, PCR/sequencing diagnosed 463 dermatophyte infections. Among these patients, fungal culture was also positive for dermatophytes in 10 (10/463, 2.2%), but was negative/contaminated in 444 (444/463, 95.7%), positive for yeasts in 4 (4/463, 0.9%), for NDMs in 5 (5/462, 1.1%) and for mixed infection in 1 (1/463, 0.2%) of these cases. These results have been visualized in Figure 1 in a simplified manner focusing on their concordance or discordance. For example, in patients with a positive dermatophyte PCR/sequencing result, culture also revealed a dermatophyte infection in 10/463 (2.2%) cases, but had a discordant result (negative/contamination, yeast, NDM or mixed infection) in 453/463 (97,8%) cases.

We specified the title in more detail and reformatted Table 2. In this way we hope that the table content becomes clearer to the readers.

Comment 1.4:  Discussion:

Clearly, the discussion needs to be improved by comparing the results with similar studies, either using 28S or using ITSs, in dermatophytes and in other fungal groups. Since the strategy used by the authors was to amplify and by sequencing 28S directly from the nail sample it should be indicated if using either 28S or ITS the results are different.  Moreover, lately there were studies using molecular biology approaches showing that sometimes the study of the rDNA is not sufficient to accurately identify dermatophytes (e.g T. rubrum complex identification). This should be included in the discussion. 

Response 1.4: Thank you very much for this thoughtful and correct comment. We previously found that the fungal identification was more sensitive using 28S DNA amplification than ITS-5S DNA amplification. Identification by 28S DNA amplification is admittedly less efficient than ITS-5S DNA to discriminate species at the species level, but sufficient to identify species at the genus level and therefore can appropriately discriminate tinea unguium on the one hand and mold onychomycosis on the other hand. The latter is crucial to choose the proper treatment for the patient. By contrast, dermatophyte identification at the species level is of less interest in onychomycosis as basically two Trichophyton spp. (T. rubrum and T. interdigitale) cause 99% of tinea unguium and their differentiation is not relevant for the choice of treatment. We have discussed this important issue in the revised manuscript (Line 281 –289).

Comment 1.5. Minor comments:

Line 164 – 166: Italics?

Line 411: Reference 40?

Response 1.5. Thank you for the careful review of our manuscript, which has been revised accordingly.

Reviewer 2 Report

The manuscript entitled “Identification of dermatophytes and non-dermatophyte agents in onychomycosis by PCR and DNA sequencing – a retrospective comparison of diagnostic tools”  (jof-1908755)is very interesting. Realy, the diagnosis of the dermatomycosis needs improvement.

The study was well conducted and the results are clear, I have some minor points to be considered:

All fungi names should be italicised, including into the tables, figures and references. On the other hand “spp.” has been written several times in italics, which is wrong, please write correctly spp. (without italizing).

I did not understand, authors claim this was a retrospective cohort study. I assumed phenotypic characterization have been achieved in the past (refs 20 -23), but  what about the DNA PCR/Sequencing? They described the method session as be unheard-of, is it correct ? In the current study what was done for the first time? Besides, if the molecular experiments have been performed posteriosly, how the nails were stored? Plese make these issues clearer.

"Other Dermatophyte" and "Other Yeast" are inappropriate in table 3, please change it

I suggest including a paragraph, in the discussion, about the limitations of the study such as low sensitivity/specificity for some fungi. As well as discussing the expected accessibility of PCR/DNA sequencing to clinical laboratories, the requirement for trained personnel, high cost, etc.

Build a paragraph with the main conclusions for the results found

Author Response

Response to Reviewer 2 Comments

The manuscript entitled “Identification of dermatophytes and non-dermatophyte agents in onychomycosis by PCR and DNA sequencing – a retrospective comparison of diagnostic tools” jof-1908755)is very interesting. Realy, the diagnosis of the dermatomycosis needs improvement.

The study was well conducted and the results are clear, I have some minor points to be considered:

Response: We wish to thank the reviewer for his positive feedback and valuable remarks. We have carefully addressed the comments of the reviewer and provided point-by-point responses as follows.

Comment 2.1: All fungi names should be italicised, including into the tables, figures and references. On the other hand “spp.” has been written several times in italics, which is wrong, please write correctly spp. (without italizing).

Response 2.1: Thank you very much for this correct remark. The manuscript has been revised according to the reviewer’s suggestions.

Comment 2.2: I did not understand, authors claim this was a retrospective cohort study. I assumed phenotypic characterization have been achieved in the past (refs 20 -23), but what about the DNA PCR/Sequencing? They described the method session as be unheard-of, is it correct? In the current study what was done for the first time? Besides, if the molecular experiments have been performed posteriosly, how the nails were stored? Plese make these issues clearer.

Response 2.2: Thank you very much for this comment. We already used and published the described technique in previous studies (e.g. Monod et al., 2006 (doi: 10.1099/jmm.0.46723-0). The detailed method section aimed to inform readers who are not familiar with all of our previous work. Application of PCR/sequencing was confined to cases with inconclusive standard diagnostics, mostly in terms of a positive fluorescence microscopic result but negative/contaminated culture. During the culture time of 2-3 weeks the remaining nail was stored unprocessed in a separate box at room temperature.  As this might be of interest for the readers, we included this information in the revised manuscript (Line 130 – 132). As we retrospectively studied the results from all nail specimens with suspected onychomycosis referred for fungal diagnostics between 2013-2018 (cohort) we think that the designation as a retrospective cohort study is appropriate.

Comment 2.3: "Other Dermatophyte" and "Other Yeast" are inappropriate in table 3, please change it

Response 2.3: The titles have been changed by removing other and by adding the umbrella term discordant result a row above. We hope that in this way the nomenclature became appropriate for the reviewer and the readers.

Comment 2.4: I suggest including a paragraph, in the discussion, about the limitations of the study such as low sensitivity/specificity for some fungi. As well as discussing the expected accessibility of PCR/DNA sequencing to clinical laboratories, the requirement for trained personnel, high cost, etc.

Response 2.4: Thank you for this comment. We already discussed some limitations of the study (missing information on antifungal pretreatment, retrospective nature and limitation of PCR/sequencing application to cases with inconclusive standard diagnostics). However, we have now build a separate and more clearly inserted paragraph for the limitations in the revised manuscript. We agree that the positivity rate of 63.5% for PCR/sequencing in microscopically positive specimens appears lower compared to other studies and we have discussed this fact in the manuscript. However, no conclusion can be drawn regarding the specifity/sensitivity for certain fungi. Furthermore, we agree that costs and the need for trained personnel needs to be considered in the assessment of molecular biology methods for the clinical routine and we have discussed this more extensively in the revised manuscript (Line 333 – 338)

Comment 2.5 Build a paragraph with the main conclusions for the results found

Response 2.5: Thank you for this suggestion. We made a separated paragraph for the conclusion and hope it is represented more clearly now.

Reviewer 3 Report

Please add reference about Fungal DNA extraction.

Please insert a discussion of costs between two approaches (culture x molecular).

Please insert the section conclusion clearly

Author Response

Response to Reviewer 3 Comments

Comment 3.1. Please add reference about Fungal DNA extraction.

Response 3.1.  Thank you for this attentive remark. We added adequate references in the revised manuscript.

Comment 3.2. Please insert a discussion of costs between two approaches (culture x molecular).

Response 3.2. We agree that costs of medical diagnostics are a relevant point to discuss and we added a brief comment on this in the discussion (Line 335 – 336)

Comment 3.3. Please insert the section conclusion clearly

Response 3.3. Thank you for this suggestion. As responded to reviewer 2, we made a separated paragraph for the conclusion and hope it is represented more clearly now.